# Measurement characteristics and correlates of HIV-related stigma among adults living with HIV: a cross-sectional study from coastal Kenya

Stanley W Wanjala ![ORCID],1,2 Moses K Nyongesa ![ORCID],3,4 Paul Mwangi,3 Agnes M Mutua,3 Stanley Luchters,1,5,6 Charles R J C Newton,3,7,8 Amina Abubakar3,5,7,8

For numbered affiliations see end of article.

**Correspondence to**
Stanley W Wanjala;
stanley.wanjala@UGent.be

## ABSTRACT

**Objective** We studied the psychometric properties of the 12-item short version of the Berger HIV stigma scale and assessed the correlates of HIV-related stigma among adults living with HIV on the Kenyan coast.

**Design** Cross-sectional study.

**Setting** Comprehensive Care and Research Centre in the Kilifi County Hospital.

**Participants** Adults living with HIV on combination antiretroviral therapy were recruited and interviewed between February and April 2018 (n=450).

**Main outcome measures** HIV-related stigma.

**Results** 450 participants with a median age of 43 years (IQR=36–50) took part in the study. Of these, 356 (79.1%) were female. Scale reliability and validity were high (alpha=0.80, test–retest reliability intraclass correlation coefficient=0.92). Using confirmatory factor analysis, we observed that the 12-item short version of the HIV stigma scale had a good fit for its hypothesised model (Comparative Fit Index=0.966, Tucker Lewis Index=0.955, root mean square error of approximation=0.044). Multigroup confirmatory factor analysis indicated measurement invariance across gender and age groups as ΔCFI was ≤0.01. Multivariate linear regression established that being female (β=2.001, 95% CI: 0.21 to 3.80, p=0.029), HIV status non-disclosure (β=4.237, 95% CI: 1.27 to 7.20, p=0.005) and co-occurrence of depressive and anxiety symptoms (β=6.670, 95% CI: 3.40 to 9.94, p<0.001) were significant predictors of perceived HIV-related stigma and that these variables accounted for 10.2% of the explained variability in HIV-related stigma among adults living with HIV from Kilifi.

**Conclusions** Our results indicate that the 12-item short version of the HIV stigma scale is a valid and reliable measure of HIV stigma in Kenya. Furthermore, our study indicates that interventions aimed at reducing stigma need to take into account gender to address the specific needs of women, people who have not disclosed their HIV status, and those exhibiting symptoms of depression and anxiety, thereby improving their quality of life.

## INTRODUCTION

HIV/AIDS remains a considerable public health concern globally, with sub-Saharan

## Strengths and limitations of this study

► This is the first study to report the 12-item HIV stigma scale's measurement characteristics in the sub-Saharan African context.

► We report on the correlates of HIV stigma based on a culturally adapted measurement tool with good psychometric properties.

► We cannot generalise our findings to all adults living with HIV in Kenya as data were collected from one geographical setting and excluded adults older than 60 years.

► We cannot conclude how individuals experience stigma over time because of the study design limitation.

Africa (SSA) bearing the most HIV-related disease burden.[1] Despite SSA making up about 11% of the earth's population, it is the world's epicentre of HIV/AIDS. By the close of 2019, an estimated 38 million people were living with HIV globally, with an estimated 68% living in SSA, accounting for two-thirds of all individuals living with HIV.[1] Between 2010 and mid-2020, there has been an upsurge in the number of people accessing antiretroviral therapy (ART) (7.8–26 million).[1] Further, between 2010 and 2019, new HIV infections declined by an estimated 16% from 2.1 million/year to 1.7 million/year, and AIDS-related deaths dropped from 1.1 million to around 690 000 per year.[1] By the end of 2019, an estimated 1.5 million Kenyans were living with HIV, with 42 000 new infections and 21 000 AIDS-related deaths reported.[2] Estimates show that between 80% and 90% of the people living with HIV/AIDS (PLWHA) in Kenya are adults.[3] Additionally, 75% of adults in Kenya are reported to be on antiretroviral treatment.[2]

Erving Goffman[4] defined stigma as a process through which individuals are 'disqualified from full social acceptance' due to an undesirable 'mark' or 'label.' This label can either be a physical, health or behavioural attribute that is regarded as 'deeply discrediting.'[4] In this study, the label is HIV seropositive status. Additionally, stigma, defined as a 'mark,' sets a person apart from others and links the person to undesirable characteristics such as stereotypes.[5] HIV-related stigma among PLWHA is prevalent throughout SSA.[6] HIV-related stigma has been identified as a severe obstacle in the way of effective responses to HIV.[7]

Although efforts have been scaled up to raise awareness and increase public knowledge about HIV since the epidemic started decades ago, social stigma is still associated with the disease.[8] Research has demonstrated that stigma keeps people from adopting HIV preventive behaviours and accessing needed care and treatment,[9] negatively impacting their health and well-being. Among women living with HIV, the decision to disclose their HIV seropositive status is likely affected by perceived stigma.[10]

From previous research, HIV stigma experienced by PLWHA can either be enacted, anticipated or internalised.[11] Enacted stigma includes an individual's experiences, prejudice, and/or discrimination from others because of one's HIV status. Anticipated stigma includes an individual's expectation of experiencing enacted stigma, while internalised stigma refers to the extent to which PLWHA have adopted negative feelings and beliefs about PLWHA.[12]

A variety of instruments designed to measure HIV-related stigma have been published.[13–21] Berger's 40-item HIV stigma scale (HSS-40) is the most commonly used instrument and one of the few instruments covering all stigma mechanisms affecting PLWHA.[12] It takes up to 25 min to complete the HSS-40,[22] which may limit its application, especially in extensive surveys. Though shortened versions covering 25[22] and 32[23] items of the HSS exist, the 12-item HSS (HSS-12)[14] version of the Berger HSS was examined in the present study as it facilitates the inclusion of HIV stigma in more extensive surveys. Furthermore, it has comparable psychometric properties to the full-length scale.[14] While evidence from other parts of the world[14] indicates that the HSS-12 is psychometrically sound, we are unaware of any study that has reported this scales' psychometric properties in the SSA context.

Empirical evidence indicates that sociodemographic characteristics such as age,[24 25] gender,[25–27] employment,[28] educational attainment[29–31] and marital status,[32] are significantly correlated with HIV-related stigma. However, the directionality is inconsistent. An explanation for the different findings regarding correlates and predictors of HIV-related stigma might be due to the diverse research strategies applied and the sample composition. Research shows that stigma and disclosure of HIV status are interrelated phenomena for PLWHA.[33] Furthermore, persons who have not disclosed their HIV status exhibit higher levels of perceived HIV-related stigma and greater levels of concern about HIV disclosure.[34]

Despite the abundance of published reports on HIV-related stigma and its predictors among specific subgroups of the adult population, there is a paucity of research findings focusing on predictors of HIV-related stigma across the entire adult population. Further, no study in the SSA context has tested for the validity and reliability of the HSS-12. This study aims to determine the correlates of HIV-related stigma among adults living with HIV from Kilifi, Coastal Kenya. Specifically, the study aims to: (1) examine the psychometric properties of the 12-item Berger Stigma Scale and (2) establish the correlates of stigma among adults living with HIV in Kilifi.

## METHODS
### Study setting
This cross-sectional study was conducted at the Kenya Medical Research Institute-Wellcome Trust Research Programme (KEMRI), Centre for Geographic Medicine Research(Coast), Kilifi, Kenya. It was based at the Comprehensive Care and Research Centre (CCRC) in the Kilifi County Hospital (KCH). The majority of Kilifi County residents are poor (71.4% live below the poverty line), lack formal education, and earn a living mainly through subsistence farming or fishing.[35–37] HIV prevalence in adults is estimated to be at 4.5%.[38] The CCRC offers clinical services such as management of opportunistic infections, HIV testing and counselling, family planning, nutritional counselling, cervical cancer screening, the dispensation of ART and serves as a research facility. About 60 patients are seen daily. By 2020, the clinic has enrolled over 9000 patients of all ages.

### Study participants
This data are part of a larger project focusing on diverse outcomes in adults living with HIV, including mental health and health-related quality of life. A cross-sectional survey of 450 study participants among patients attending an HIV care and treatment clinic at the KCH was conducted between February and April 2018 (figure 1). The participation criteria were age (18–60 years old) with confirmed HIV positive status, on combination ART (cART), and informed consent to participate. Participants with an acute medical illness or cognitive difficulties at the time of enrolment/administration of questionnaire or could not understand and/or communicate in the national language (Kiswahili), which was used during the administration of all study instruments, were excluded. A research team member introduced the study to eligible participants when they visited the clinic for scheduled appointments. Those who consented to take part responded to the instruments at the clinic.

### Data collection procedures
Study data were collected and managed using Research Electronic Data Capture (REDCap) tools hosted at

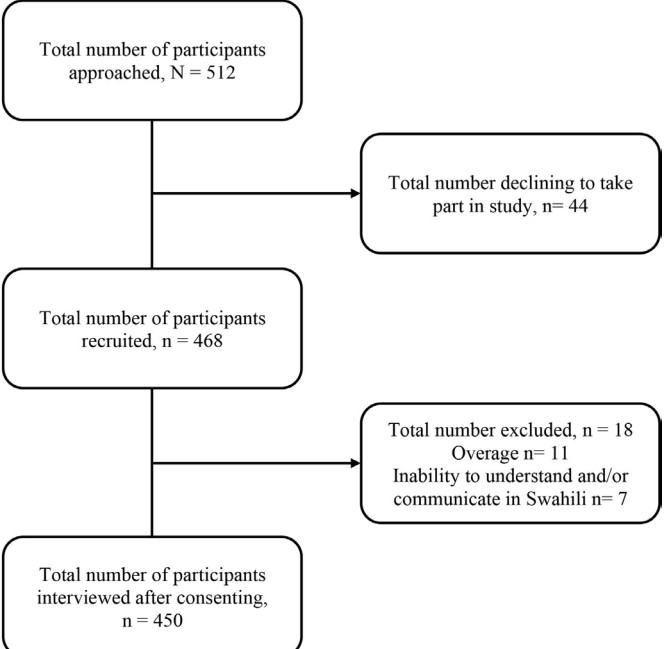

**Figure 1** Study recruitment flow chart.

KEMRI Wellcome Trust Programme.[39 40] REDCap is a secure, web-based software platform designed to support data capture for research studies, providing (1) an intuitive interface for validated data capture; (2) audit trails for tracking data manipulation and export procedures; (3) automated export procedures for seamless data downloads to common statistical packages and (4) procedures for data integration and interoperability with external sources. Data collection instruments were interviewer-administered via android tablets, in the same order and under the same administration environment. Research assistants underwent a 4-day training in research ethics and proper interviewing techniques (with role-plays) and were familiarised with the tablet-based questionnaires. The questionnaire administration took place in a quiet and private room within the CCRC in KCH, and the interview session lasted between 30 and 45 min.

### Measures
#### HIV-related stigma
The short version (HSS-12) of the Berger HSS[14] was used to assess patient-perceived HIV-related stigma under four dimensions: (1) personalised stigma; (2) disclosure concerns; (3) negative self-image and (4) concerns with public attitudes, each comprising a subscale of the instrument. Personalised stigma has been suggested to represent the enacted stigma mechanism, disclosure concerns, and concerns with public attitudes dimensions have been proposed to represent anticipated stigma mechanism, and negative self-image has been proposed to represent internalised stigma mechanism.[12] Items on this scale are rated from 1 to 4, with (1) being 'strongly disagree' and (4) 'strongly agree.' The possible score for each item ranges from 1 to 4 (3–12 for subscale), and a total score ranges between 12 and 48 and is derived from the summation of item scores. Higher scores designate a greater level of perceived HIV-related stigma.

#### Patient Health Questionnaire version 9
Patient Health Questionnaire version 9 (PHQ-9)[41] was administered as a measure of depressive symptoms. The PHQ-9 is a nine-item scale rated on a Likert-type scale ranging from 0 'not at all' to 3 'nearly every day.' Item scores are summated to derive a total score ranging from 0 to 27. It has previously been found to have good internal consistency (Cronbach alpha 0.78) and acceptable test–retest reliability (intraclass correlation coefficient (ICC)=0.59) when used among adults living with HIV infection in Kenya.[42]

#### Generalised Anxiety Disorder version 7
Generalised Anxiety Disorder version 7 (GAD-7)[43] was administered as a clinical measure for assessing GAD based on the Diagnostic and Statistical Manual of Mental Disorders (DSM-IV) criteria. The GAD-7 is a seven-item self-report instrument rated on a Likert-type scale ranging from 0 'not at all' to 3 'nearly every day.' The scale score ranges from 0 to 21. There is evidence in support of the reliability and validity of this scale in Kenya.[44] Scores from PHQ-9 and GAD-7 were combined to generate a variable called symptoms of common mental disorders comorbidity, indicating the co-occurrence of depressive and anxiety symptoms.

#### Sociodemographic and asset index items
A sociodemographic questionnaire was used to collect information on the participants' age, gender, relationship status, educational level, employment status and whom they currently shared a residence. Furthermore, an asset index previously used in this setting[45] was used to collect information about participants' socioeconomic status (SES) based on disposable assets owned. Participants were asked for ownership of disposable items such as radio, television, refrigerator, gas, bicycle, motorcycle and car. The final SES score had seven (7) items. A total asset score is calculated, and higher scores indicate a better SES. The maximum possible score for the asset index score was 7. An asset index to estimate family wealth has been recommended as an alternative approach to estimating SES in settings where reliable data on family income may not be available.[46]

#### Clinical information
Participants' data were extracted from the clinic's medical record database and filled into a clinical record form. This information included participants' dates of HIV-diagnosis, cART initiation, most current cART regimen, cluster of differentiation 4 (CD4) cell count, viral load (within the last 1 year), recent height and weight (for body mass index (BMI) calculation) and data on WHO clinical staging. Participants' clinical information was retrieved from their clinical records after consent was granted. Patient-unique clinic numbers were used to access participants' medical records. We report substantial missing

participant data on viral load from the database (n=145) with no follow-up record of CD4 cell count for all study participants.

## Instrument translation and cross-cultural adaptation

The English version of the HSS-12 was forward translated by two independent bilingual translators to Kiswahili and back-translated into English by two independent back translators (oblivious of the original version). A group of Kenyan HIV researchers bilingual and fluent in both Kiswahili and English and the translators had a harmonisation meeting to review the content, conceptual, semantic and idiomatic equivalence of the questionnaires to ensure the cultural relevance of the HSS-12. Before conducting the formal phase of the study, 15 pretest interviews were conducted to assess instrumentation rigour and develop measures to address any limitations or threats to bias and management procedures. The final version of the questionnaire was obtained after the incorporation of changes emerging from pretesting. Pretesting procedures have been elaborated further elsewhere.[47]

## Patient and public involvement

Patients were not involved in the design and conduct of this study.

## Statistical analyses

### Factor structure and measurement invariance across age groups and gender

First, confirmatory factor analysis (CFA) was used to examine the HSS's factor structure. A CFA model representing the Swahili version of the HSS-12 was set up and analysed with weighted least square mean and variance adjusted using the lavaan[48] package in R statistical software[49] on all the 450 observations. The Goodness of fit was assessed using the $\chi^2$ test, Comparative Fit Index (CFI), Tucker Lewis Index (TLI) and root mean square error of approximation (RMSEA). The data were expected to have a good fit to the model if the $\chi^2$ test was non-significant, CFI and TLI values were greater than 0.90, and RMSEA score was lower than 0.05.[50]

Second, after defining the model, Multi-Group CFA[51] was used to test for measurement invariance of the HSS-12 for gender and age groups. Change in CFI (ΔCFI) has been suggested as a robust statistic for testing the between-group invariance of CFA models. Additionally, it has been recommended that invariance can be assumed when ΔCFI is ≤0.01 in absolute values.[52]

### Internal construct validity and convergent validity

Means and SD were used to evaluate the distribution of scores within the subscale and among the items. Itemised means and SD were expected to be almost the same within the subscale, justifying item scores' aggregation into subscale scores.[53] The item-total correlation was used to evaluate internal construct validity. Each items' corrected item-total correlation coefficients were calculated and expected to exceed 0.4 and vary in range. Convergent validity was assessed using the Pearson correlation

coefficient between HSS-12, PHQ-9, and GAD-7 scores. Correlation coefficients were interpreted as small (0.10–0.29), moderate (0.30–0.49) and large (0.49 and above).[54]

### Reliability

Cronbach's alpha and ordinal alpha were used to examine each subscale's internal consistency and overall scores of the Swahili version of the HSS-12. Cronbach's alpha was considered acceptable if greater than (>0.7).[55] The ICC was used to examine test–retest of the Swahili version of the HSS-12 by correlating scores taken at two different time points (2 weeks apart)[56] using the same measure administered to the same participant. ICC of 0.60 was considered marginal, 0.70 acceptable and anything over 0.80 considered high.[57]

## Sample characteristics and correlates

Frequencies and means (with percentages and SD) were used to describe sample characteristics. Univariate and multivariable linear regression were used to assess factors associated with both stigma subscales and the overall stigma scale. In the regression model, stigma scores were expressed as a continuous measure. Independent variables included age, gender, marital status, education level, employment status, SES, BMI, viral load, WHO clinical stages, months since HIV diagnosis, months since cART initiation, HIV status disclosure, self-reported opportunistic infections, and the co-occurrence of depressive and anxiety symptoms. Our review of the literature informed factors included in the model. All variables with p<0.20 were included in the multivariable regression model apart from viral load because participants had missing values (n=145). The final multivariable models were generated using a backward stepwise approach by eliminating all variables independently with p>0.05. Assumptions of linear regression testing were visually inspected through histograms (linearity), normal probability plots (normality) and plots of residual vs predicted values (homoscedasticity). Multicollinearity was assessed using the variance inflation factor. There were no multicollinearity problems. Modelling was undertaken five times in total: once to predict overall stigma and once to predict each of the four subscales. R (V.4.0.2) statistical software package[49] was used to explore the construct validity of the HSS-12. All other analyses were run using (Stata V.14.0) statistical software package.[58]

## RESULTS

### Sample characteristics

The 450 participants had a median age of 43 years (IQR=36–50), ranging from 18 to 60 years. The vast majority of the sample were female (79.1%), had attained basic primary level education (53.1%), lived with a family member (82.4%) and were unemployed (59.8%). Less than half of the study participants (43.8%) were separated, divorced or widowed. The mean BMI was within the normal range (mean (SD)=22.4 (4.8)). Most study

participants had disclosed their HIV status to others (94.0%). The median time since HIV diagnosis was 8.8 years (IQR=4.67–11.50), ranging from 0 to 18 years. A total of 417 (93.7%) were in stage 1 of the WHO clinical staging, and 425 (95.3%) were on the first-line cART regimen (table 1). The median time elapsed since cART initiation was 6.7 years (IQR=3.67–10.00). At the time of the interview, less than a fifth (18.4%) of the study participants had an opportunistic infection.

Perceived overall stigma scores ranged from 12 to 48, with a median score of 28 (IQR=23–33). Using PHQ-9 and GAD-7 cut-off score of ≥10, which has been shown to maximise specificity and sensitivity for depression[59] and GAD[43] screening, the overall prevalence of depression and anxiety was 13.8% and 5.3%, respectively, among enrolled participants. The co-occurrence of depressive and anxiety symptoms was present in 4.7% of the study participants.

### Factor structure and measurement invariance across age groups and gender

Online supplemental figure 1 presents CFA results with standardised correlation coefficients. Our hypothesised model that the overall stigma scale comprises four subscales correlated was confirmed given the observed fit indexes. The $\chi^2$ test was statistically significant ($\chi^2$=91.982, df=50, p=0.000) but alternate fit measures indicated acceptable fit; RMSEA: 0.044; CFI:0.966 and TLI: 0.955. These results generally indicate that the data had a good fit to the model and that we can confidently use both total and subscale scores in this population. Measurement invariance across age groups and gender was supported because ΔCFIs are lower than 0.01 in all models suggesting that measurement invariance can be assumed.

### Internal construct validity and convergent validity

The factor loading of all items on the hypothesised scale was good except for item 6 (0.21) under the disclosure concern subscale. Convergent validity of the HSS-12 was demonstrated by the small to moderate correlations between HSS-12 and the correlation with the following relevant measures: GAD-7 (r=0.368, p<0.001) and PHQ-9 (r=0.328, p<0.001) table 2.

### Reliability: internal consistency and test–retest

Cronbach's alpha (α) for the subscales and overall scale were all >0.7 (see table 2) except for the disclosure concern sub-scale, which was 0.53 (95% CI: 0.46 to 0.60). Additionally, ordinal α for the subscales ranged from 0.65 to 0.91. The test–retest reliability of the short 12-item version of the HSS was excellent, 0.92 (95% CI: 0.87 to 0.95). Additionally, table 2 presents descriptive statistics for the stigma scale on the item level and subscale level. Corrected item-total correlation coefficients were >0.4 for all the items apart from one item (0.21) in the disclosure concerns subscale. A variation of 0.46–0.88 indicates

| | Total sample | |
|---|---|---|
| **Sample characteristics** | **N=450** | **%** |
| Sociodemographic characteristics | | |
| Age—years range (18–60), median (IQR) | 43 (14) | |
| Gender | | |
| Female | 356 | 79.1 |
| Male | 94 | 20.9 |
| Marital status | | |
| Married/cohabiting | 196 | 43.6 |
| Separated/divorced/widowed | 197 | 43.8 |
| Single/never married | 57 | 12.7 |
| Education | | |
| Tertiary | 22 | 4.9 |
| Secondary | 66 | 14.7 |
| Primary | 239 | 53.1 |
| None | 123 | 27.3 |
| Employment | | |
| Formally employed | 53 | 11.8 |
| Self-employed | 117 | 26.0 |
| Other | 11 | 2.4 |
| Unemployed (including students) | 269 | 59.8 |
| Currently living with | | |
| Family | 371 | 82.4 |
| Relative/friend | 10 | 2.2 |
| Alone | 69 | 15.3 |
| Asset index score*—mean (SD) | 1.2 (1.4) | |
| Perceived HIV-stigma score†—mean (SD) | 28.4 (7.7) | |
| Any current chronic illness | | |
| No | 413 | 91.8 |
| Yes | 37 | 8.2 |
| Clinical characteristics | | |
| BMI—kg/m², mean (SD), OM=4 | 22.4 (4.8) | |
| cART regimen, OM=4 | | |
| First line | 425 | 95.3 |
| Second line | 21 | 4.7 |
| Viral load, OM=145 | | |
| ≤1000 copies/mL | 265 | 86.9 |
| >1000 copies/mL | 40 | 13.1 |
| WHO clinical stage, OM=5 | | |
| Stage 1 | 417 | 93.7 |
| Stage 2 | 22 | 4.9 |

Continued

| | **Total sample** | |
|---|---|---|
| **Sample characteristics** | **N=450** | **%** |
| Stage 3 | 3 | 0.7 |
| Stage 4 | 3 | 0.7 |
| Months since HIV diagnosis— median (IQR) | 106 (82) | |
| Months since cART initiation— median (IQR) | 80.5 (76) | |
| Treatment characteristics | | |
| HIV status disclosure | | |
| Yes | 423 | 94.0 |
| No | 27 | 6.0 |
| Any current opportunistic infection | | |
| No | 367 | 81.6 |
| Yes | 83 | 18.4 |

**Table 1** Continued

*Score range = 0–7.
†Score range = 12–48.
BMI, body mass index; cART, combination antiretroviral therapy; IQR, Interquartile range; OM, observation with missing value; WHO, World Health Organization.

that the intended stigma concepts' broadness had been captured.

### Correlates of perceived HIV-related stigma

Tables 3 and 4 present results based on univariate and multivariable regression analyses, respectively. In the univariate model, it was found that being female, being separated, divorced or widowed, having primary or no level of education, being self-employed or unemployed, having a low asset index score, having a viral load of >1000 copies/mL, decreased duration since HIV diagnosis, decreased duration since cART initiation, HIV status non-disclosure, having any current opportunistic infection and co-occurrence of depression and anxiety symptoms were significantly associated with overall HIV stigma scores.

Personalised stigma was significantly associated with being female, being single, separated, divorced or widowed, self-employed or unemployed, having a low asset index score, having a viral load of >1000 copies/mL, having any current opportunistic infection, and the co-occurrence of depressive and anxiety symptoms. Disclosure concern was significantly associated with being separated, divorced or widowed, having no level of education, having a low asset index score, less time elapsed since HIV diagnosis, less time elapsed since cART initiation, and HIV status non-disclosure. Concern with public attitudes was significantly associated with being female, having primary or no level of education, decreased duration since cART initiation, and the co-occurrence of depressive and anxiety symptoms. Negative self-image was significantly associated with being separated, widowed or divorced, having no

level of education, being self-employed or unemployed, having a viral load of >1000 copies/mL, decreased duration since HIV diagnosis, decreased duration since cART initiation, having any current opportunistic infection and the co-occurrence of depressive and anxiety symptoms.

When a multiple linear regression model was run, it was found that being female ($\beta$=2.001, 95% CI: 0.21 to 3.80, p=0.029), HIV status disclosure ($\beta$=4.237, 95% CI: 1.27 to 7.20, p=0.005) and co-occurrence of depressive and anxiety symptoms ($\beta$=6.670, 95% CI: 3.40 to 9.94, p<0.001) were significant predictors of perceived HIV stigma. Having no education was associated with increasing stigma levels at p=0.051 ($\beta$=3.318, 95% CI: −0.01 to 6.65). Regression results indicated that the model explained 10.2% of the variance and that the model was a significant predictor of perceived HIV stigma $F_{(6, 395)}$=7.46, p<0.001).

Concerning the four subscales, we found that personalised stigma was positively correlated with being female and the co-occurrence of depressive and anxiety symptoms. Disclosure concern was inversely correlated with duration since HIV diagnosis and positively correlated with having no level of education and HIV status non-disclosure. Concerns with public attitudes were positively correlated with being female. Negative self-image was positively correlated with having no level of education and the co-occurrence of depressive and anxiety symptoms.

## DISCUSSION

This cross-sectional analysis of data from adults living with HIV observed that the HSS-12 presents excellent psychometric properties. Additionally, we observed that stigma was associated with both physical and mental well-being. According to our study, correlates of HIV-related stigma include being female, HIV status non-disclosure, and the co-occurrence of depressive and anxiety symptoms.

### Factor structure, measurement invariance, validity and reliability of the short 12-item Swahili version of the HSS

The study examined the stigma scale's psychometric properties to assess its usefulness and describe the correlates of HIV-related stigma among adults living with HIV in Kilifi. Reliability and validity were acceptable, and CFA supported the four-factor solution measuring the four dimensions of HIV stigma. Cronbach's alpha for the HSS-12 among the Kenyan population is similar to the Swedish population in which the scale was developed.[14] Although Cronbach's alpha for the adapted HSS-12 subscales was slightly lower (0.53–0.84) than the initial version of HSS-12 (0.80–0.88), its' alpha for the total scale was 0.80 suggesting good internal consistency. Furthermore, the adapted HSS-12 had an ordinal alpha of 0.86. The difference between ordinal alpha and Cronbach's alpha values could be attributed to high skewness and kurtosis values for some of the questionnaire's questions, influencing Cronbach's alpha estimate values.[60 61]

Measurement invariance of the Swahili HSS-12 was evaluated and confirmed across main interest groups:

**Table 2** Descriptive statistics for items and subscales in the short form 12-item Swahili version of the HIV stigma scale

| Item | Mean item score† (SD) | Corrected item correlation | Mean subscale score‡ (SD) | Reliability | | Validity | | | |
|---|---|---|---|---|---|---|---|---|---|
| | | | | Internal consistency (Cronbach α) | Test-retest (ICC) | Convergent | | Construct | |
| | | | | | | § | ¶ | CFI | RMSEA | TLI |
| Personalised stigma | | | 4.86 (2.56) | 0.84 (95% CI 0.81 to 0.86) | 0.83 (95% CI 0.71 to 0.90) | 0.357* | 0.327* | | |
| Some people stop touching me soon they know/realise I am infected with HIV/AIDS. | 1.66 (1.01) | 0.65 | | | | | | | |
| People I care for stopped calling me after knowing I suffer from AIDs. | 1.63 (1.00) | 0.87 | | | | | | | |
| I have lost friends for telling/explaining that I have AIDs. | 1.59 (0.96) | 0.88 | | | | | | | |
| Disclosure concerns | | | 8.74 (2.37) | 0.53 (95% CI 0.45 to 0.60) | 0.62 (95% CI 0.36 to 0.77) | 0.070 | 0.070 | | |
| Telling someone that I have AIDs is dangerous.† | 2.24 (1.24) | 0.83 | | | | | | | |
| I do all I can to keep my AIDs (HIV) status secret. | 2.90 (1.22) | 0.46 | | | | | | | |
| I am very careful to that person I tell about my HIV status (I am cautious/very cautious/very careful to (?of) the people I tell my HIV status). | 3.60 (0.78) | 0.21 | | | | | | | |
| Concerns about public attitudes | | | 8.52 (3.17) | 0.83 (95% CI 0.80 to 0.86) | 0.79 (95% CI 0.65 to 0.88) | 0.187* | 0.165* | | |
| People who are suffering from AIDs are treated as if they are not like the other people. | 3.05 (1.18) | 0.68 | | | | | | | |
| People believe that a person infected with HIV is dirty. | 2.74 (1.26) | 0.84 | | | | | | | |
| Many people are worried when they are near a person infected with HIV. | 2.75 (1.22) | 0.84 | | | | | | | |
| Negative self-image | | | 6.32 (3.00) | 0.74 (95% CI 0.70 to 0.80) | 0.76 (95% CI 0.60 to 0.86) | 0.372* | 0.330* | | |
| I feel guilty because I am infected with HIV. | 2.11 (1.23) | 0.60 | | | | | | | |
| People's attitudes about HIV/AIDs makes me feel very bad. | 2.23 (1.25) | 0.78 | | | | | | | |
| I feel I am not as good as others because I'm infected with HIV. | 2.01 (1.23) | 0.73 | | | | | | | |
| Overall | | | 28.44 (7.68) | 0.80 (95% CI 0.77 to 0.83) | 0.92 (95% CI 0.87 to 0.95) | 0.368* | 0.328* | 0.966 | 0.044 | 0.955 |

Pearson product-moment correlation coefficient.
*p<0.001
†Possible score for each item 1–4; higher scores reflect a higher level of perceived HIV-related stigma
‡Possible score 3–12 on each sub-scale; higher scores reflect a higher level of perceived HIV-related stigma.
§correlation between HIV stigma and GAD-7.
¶correlation between HIV stigma and PHQ-9.
CFI, Comparative Fit Index; Cronbach α, Cronbach alpha; RMSEA, root mean square error of approximation; SD, Standard Deviation; TLI, Tucker Lewis Index.

**Table 3** Univariate linear regression of correlates of perceived HIV-related stigma among adults living with HIV from rural Kilifi

| Independent variables | N | Personalised stigma B (95% CI) | P value | Disclosure concerns B (95% CI) | P value | Public attitudes B (95% CI) | P value | Negative self-image B (95% CI) | P value | Overall HIV stigma score† B (95% CI) | P value |
|---|---|---|---|---|---|---|---|---|---|---|---|
| Sociodemographic characteristics | | | | | | | | | | | |
| Age | 450 | −0.01 (−0.03 to 0.02) | 0.595 | 0.01 (−0.02 to 0.02) | 0.999 | 0.01 (−0.02 to 0.04) | 0.399 | −0.01 (−0.03 to 0.03) | 0.880 | 0.01 (−0.07 to 0.08) | 0.910 |
| Gender | 450 | | | | | | | | | | |
| Male | | Ref | | Ref | | Ref | | Ref | | Ref | |
| Female | | 0.52 (−0.06 to 1.10) | 0.080* | 0.29 (−0.25 to 0.83) | 0.074 | 1.07 (0.35 to 1.79) | 0.003** | 0.40 (−0.29 to 1.08) | 0.255 | 2.27 (0.54 to 4.01) | 0.010** |
| Marital status | 450 | | | | | | | | | | |
| Married | | Ref | | Ref | | Ref | | Ref | | Ref | |
| Separated/divorced/widowed | | 0.54 (0.03 to 1.04) | 0.038** | 0.67 (0.20 to 1.14) | 0.005** | 0.25 (−0.38 to 0.87) | 0.442 | 0.73 (0.14 to 1.32) | 0.016** | 2.18 (0.67 to 3.69) | 0.005** |
| Single/never married | | 0.61 (−0.14 to 1.37) | 0.111* | 0.17 (−0.52 to 0.87) | 0.626 | −0.43 (−1.37 to −0.51) | 0.369 | 0.40 (−0.49 to 1.28) | 0.378 | 0.75 (−1.50 to 3.01) | 0.512 |
| Education level | 450 | | | | | | | | | | |
| Tertiary | | Ref | | Ref | | Ref | | Ref | | Ref | |
| Secondary | | −0.12 (−1.36 to 1.12) | 0.847 | 0.08 (−1.06 to 1.21) | 0.896 | 0.68 (−0.84 to 2.21) | 0.380 | −0.03 (−1.46 to 1.40) | 0.967 | 0.61 (−3.05 to 4.26) | 0.745 |
| Primary | | −0.31 (−1.43 to 0.81) | 0.582 | 0.48 (−0.55 to 1.51) | 0.360 | 1.32 (−0.06 to 2.70) | 0.061* | 0.72 (−0.57 to 2.01) | 0.273 | 2.20 (−1.10 to 5.51) | 0.191* |
| None | | 0.15 (−1.01 to 1.32) | 0.794 | 1.23 (0.16 to 2.30) | 0.024** | 1.80 (0.36 to 3.23) | 0.014** | 1.63 (0.29 to 2.97) | 0.018** | 4.81 (1.38 to 8.25) | 0.006** |
| Employment status | 450 | | | | | | | | | | |
| Formally employed | | Ref | | Ref | | Ref | | Ref | | Ref | |
| Self-employed | | 0.67 (−0.16 to 1.50) | 0.112* | 0.27 (−0.50 to 1.05) | 0.490 | 0.46 (−0.57 to 1.49) | 0.385 | 0.73* (−0.24 to 1.70) | 0.141* | 2.13 (−0.36 to 4.62) | 0.094* |
| Other | | −0.67 (−2.33 to 0.99) | 0.429 | −0.02 (−1.57 to 1.53) | 0.983 | −1.14 (−3.20 to 0.93) | 0.279 | −0.35 (−2.29 to 1.60) | 0.726 | −2.17 (−7.15 to 2.81) | 0.392 |
| Unemployed | | 0.51 (−0.25 to 1.26) | 0.187* | 0.33 (−0.37 to 1.03) | 0.360 | 0.18 (−0.76 to 1.11) | 0.710 | 1.03 (0.15 to 1.91) | 0.022** | 2.04 (−0.22 to 4.30) | 0.077* |
| Currently living with | 450 | | | | | | | | | | |
| Immediate family | | Ref | | Ref | | Ref | | Ref | | Ref | |
| Relative/friend | | 0.86 (−0.75 to 2.47) | 0.294 | 0.02 (−1.47 to 1.52) | 0.975 | 0.18 (−1.82 to 2.18) | 0.862 | −0.45 (−2.34 to 1.45) | 0.644 | 0.62 (−4.23 to 5.46) | 0.802 |
| Alone | | 0.01 (−0.65 to 0.66) | 0.995 | −0.25 (−0.87 to 0.36) | 0.414 | −0.06 (−0.88 to 0.76) | 0.887 | −0.07 (−0.84 to 0.70) | 0.860 | −0.38 (−2.36 to 1.60) | 0.706 |
| Asset index score†—mean (SD) | 450 | −0.12 (−0.29 to 0.05) | 0.171* | −0.13 (−0.29 to 0.03) | 0.109* | −0.11 (−0.33 to −0.10) | 0.310 | −0.12 (−0.32 to −0.08) | 0.244* | −0.48 (−1.00 to −0.04) | 0.068* |
| Clinical characteristics | | | | | | | | | | | |
| BMI—kg/m², mean (SD); OM=4 | 450 | 0.004 (−0.04 to 0.05) | 0.855 | −0.03 (−0.07 to 0.02) | 0.244 | 0.03 (−0.03 to 0.09) | 0.309 | −0.03 (−0.18 to 0.12) | 0.708 | −0.03 (−0.18 to 0.12) | 0.708 |
| Viral load OM=145 | 305 | | | | | | | | | | |
| ≤1000 copies/mL | | Ref | | Ref | | Ref | | Ref | | Ref | |
| >1000 copies/mL | | 0.58 (−0.28 to 1.44) | 0.183* | 0.10 (−0.70 to 0.90) | | 0.07 (−1.00 to 1.14) | 0.894 | 1.05 (0.08 to 2.02) | 0.033** | 1.81 (−0.79 to 4.40) | 0.172* |
| Months since HIV diagnosis | 450 | 0.00 (−0.00 to 0.01) | 0.346 | −0.01 (−0.01 to −0.00) | 0.001** | −0.00 (−0.01 to 0.00) | 0.630 | −0.01 (−0.01 to 0.00) | 0.057* | −0.01 (0.03 to 0.00) | 0.091* |
| Months since cART initiation OM=4 | 446 | 0.00 (−0.00 to 0.01) | 0.497 | −0.01 (−0.01 to −0.00) | 0.001*** | −0.00 (−0.01 to 0.00) | 0.202* | −0.01 (−0.01 to −0.00) | 0.031** | −0.02 (−0.03 to −0.00) | 0.031** |
| Treatment characteristics | | | | | | | | | | | |
| HIV status disclosure | 450 | | | | | | | | | | |
| Yes | | Ref | | Ref | | Ref | | Ref | | Ref | |

Continued

**Table 3** Continued

| Independent variables | N | Personalised stigma B (95% CI) | P value | Disclosure concerns B (95% CI) | P value | Public attitudes B (95% CI) | P value | Negative self-image B (95% CI) | P value | Overall HIV stigma score‡ B (95% CI) | P value |
|---|---|---|---|---|---|---|---|---|---|---|---|
| No | | 0.23 (−0.77 to 1.23) | 0.651 | 1.86 (0.94 to 2.77) | 0.000*** | 0.67 (−0.57 to 1.91) | 0.287 | 0.72 (−0.45 to 1.89) | 0.228 | 3.47 (0.49 to 6.46) | 0.022** |
| Any current opportunistic infections | 450 | | | | | | | | | | |
| No | | Ref | | Ref | | Ref | | Ref | | Ref | |
| Yes | | 0.65 (0.04 to 1.26) | 0.037** | 0.09 (−0.48 to 0.65) | 0.786 | 0.12 (−0.64 to 0.88) | 0.759 | 0.87 (0.16 to 1.59) | 0.017** | 1.72 (−0.11 to 3.55) | 0.065* |
| CMD comorbidity OM=48 | 402 | | | | | | | | | | |
| Absence | | Ref | | Ref | | Ref | | Ref | | Ref | |
| Presence | | 2.71 (1.58 to 3.84) | 0.000*** | 0.18 (−0.91 to 1.28) | 0.741 | 1.09 (−0.38 to 2.55) | 0.144* | 3.07 (1.76 to 4.39) | 0.000*** | 7.06 (3.71 to 10.41) | 0.000*** |

Overall stigma scale represents the sum of all twelve items from the four subscales; A negative stigma score indicates less stigma.
*P<0.20, **p<0.05, ***p<0.001.
†Score range=0–7.
‡Score range=12–48.
BMI, body mass index; cART, combination antiretroviral therapy; CMD, symptoms of depression and anxiety; OM, observation with missing value; Ref, reference category.

gender and age. Our results indicated that the measurement model of the Swahili HSS-12 as a patient-reported outcome to measure perceived HIV stigma among adults is comparable across age groups and gender (table 5).

Test–retest reliability, an indicator of scale stability over time, was of acceptable levels. The original HSS-40 has been used in diverse settings[13 62] among adults 18 years and above, reporting a test retest reliability between (ICC=0.89–0.92). To the best of our knowledge, no study has reported the test retest reliability of the HSS-12.

We examined the construct validity of the scale using CFA since its hypothesised structure has been published.[14] Our results indicated that the hypothesised model fit the data well and was almost similar to what was reported by a study conducted in Sweden.[14] These results indicate that one can use both the total scores and the subscale scores and interpret the results in confidence, knowing that the items fit well together. HSS-12 evidenced convergent validity by being correlated with PHQ-9, a measure of depression and GAD-7, a measure of anxiety in conventional ways.

The HSS-12 was reliable and valid for detecting stigma among adults living with HIV at the Kenyan Coast. Consequently, HSS-12 can be practically used as a brief screening tool for stigma-related problems both for research and clinical purposes. Future research could examine its predictive validity and evaluate its sensitivity to changes. This information would be crucial in determining its usefulness as an evaluation tool for programmes and interventions.

## Correlates of stigma

Being female was positively associated with increased perceived HIV-related stigma scores, personalised stigma, and concern with public attitudes. This finding agrees with previous studies from SSA[63] and outside[28 64] that reported a positive association between female gender and perceived HIV-related stigma. Research shows that females are more likely to suffer from stigma in patriarchal societies like Kenya than males.[65 66] Research has established that the African society is less tolerant of females living with HIV than males living with HIV.[67 68] Due to women's subordinate status in society, they are often stigmatised as vectors of transmission.[69] Furthermore, the common belief that HIV is caused by indecent sexual behaviour has worse societal consequences for women who are expected to be monogamous, unlike men in most African societies.[67] Women are often blamed counterfactually to be responsible for HIV transmission.[67] Similar processes can be assumed to be at work in the Kenyan coastal region.

HIV status disclosure was positively associated with overall HIV-related stigma scores and disclosure concerns, with persons who had not disclosed their HIV status reporting greater levels of concern about HIV disclosure concerns. Anakwa *et al* found that PLWHA with higher levels of perceived HIV-related stigma reported greater levels of HIV disclosure concerns; therefore, they are less likely to disclose

**Table 4** Multivariate linear regression of correlates of perceived HIV-related stigma among adults living with HIV from rural Kilifi

| | Dependent variables | | | | | | | | | |
| --- | --- | --- | --- | --- | --- | --- | --- | --- | --- | --- |
| | Personalised stigma (n=402) | | Disclosure concerns (n=450) | | Public attitudes (n=450) | | Negative self-image (n=402) | | Overall HIV Stigma Score (n=402) | |
| Independent variables | B (95% CI) | P value | B (95% CI) | P value | B (95% CI) | P value | B (95% CI) | P value | B (95% CI) | P value |
| Sociodemographic characteristics | | | | | | | | | | |
| Gender | | | | | | | | | | |
| Male | Ref | | | | Ref | | | | Ref | |
| Female | 0.75 (0.17 to 1.34) | 0.012** | | | 1.07 (0.35 to 1.79) | 0.003** | | | 2.00 (0.21 to 3.80) | 0.029** |
| Education level | | | | | | | | | | |
| Tertiary | | | Ref | | | | Ref | | Ref | |
| Secondary | | | −0.04 (−1.14 to 1.07) | 0.950 | | | −0.05 (−1.44 to 1.33) | 0.939 | −0.34 (−3.83 to 3.16) | 0.850 |
| Primary | | | 0.48 (−0.52 to 1.48) | 0.346 | | | 0.51 (−0.73 to 1.74) | 0.423 | 1.37 (−1.75 to 4.50) | 0.388 |
| None | | | 1.24 (0.20 to 2.28) | 0.019** | | | 1.33 (0.04 to 2.62) | 0.044** | 3.32 (−0.01 to 6.65) | 0.051 |
| Clinical characteristics | | | | | | | | | | |
| Months since HIV diagnosis | | | −0.01 (−0.01 to 0.00) | 0.007** | | | | | | |
| Treatment characteristics | | | | | | | | | | |
| HIV status disclosure | | | | | | | | | | |
| Yes | | | Ref | | | | | | Ref | |
| No | | | 1.79 (0.88 to 2.70) | 0.000*** | | | | | 4.24 (1.27 to 7.20) | 0.005** |
| CMD comorbidity | | | | | | | | | | |
| Absence | Ref | | | | | | Ref | | Ref | |
| Presence | 2.67 (1.55 to 3.79) | 0.000*** | | | | | 3.04 (1.74 to 4.34) | 0.000*** | 6.67 (3.40 to 9.94) | 0.000*** |
| Variance explained by the model Pseudo R-squared | 6.76% | | 8.66% | | 1.89% | | 7.71% | | 10.17% | |

Overall stigma scale represents the sum of all twelve items from the four subscales.
**P<0.05, ***p<0.001.
CI, Confidence Interval; CMD, symptoms of depression and anxiety; Ref, Reference category.

**Table 5** Multigroup confirmatory factor analysis for age and gender subgroups

| Invariance steps | Gender | RMSEA | TLI | CFI | ΔCFI | Age | RMSEA | TLI | CFI | ΔCFI |
|---|---|---|---|---|---|---|---|---|---|---|
| Configural invariance | Female | 0.051 | 0.934 | 0.950 | | Older adults | 0.040 | 0.960 | 0.970 | |
| | Male | 0.051 | 0.934 | 0.950 | | Young adults | 0.040 | 0.960 | 0.970 | |
| Metric invariance | Female | 0.052 | 0.932 | 0.943 | 0.007 | Older adults | 0.042 | 0.957 | 0.964 | 0.006 |
| | Male | 0.052 | 0.932 | 0.943 | 0.007 | Young adults | 0.042 | 0.957 | 0.964 | 0.006 |
| Scalar invariance | Female | 0.050 | 0.936 | 0.943 | 0.000 | Older adults | 0.041 | 0.959 | 0.963 | 0.001 |
| | Male | 0.050 | 0.936 | 0.943 | 0.000 | Young adults | 0.041 | 0.959 | 0.963 | 0.001 |
| Strict invariance | Female | 0.048 | 0.941 | 0.942 | 0.001 | Older adults | 0.041 | 0.959 | 0.960 | 0.003 |
| | Male | 0.048 | 0.941 | 0.942 | 0.001 | Young adults | 0.041 | 0.959 | 0.960 | 0.003 |

Criteria for an acceptable fit were a RMSEA of <0.06, and a CFI, and a TLI of ≥0.90. Configural invariance—no constraints; Full metric invariance—with all factor loadings constrained equal. Scalar invariance—with all intercepts constrained equal; Strict invariance—with all factor loadings and intercepts fixed; Measurement invariance is assumed when ΔCFI is ≤0.01.
CFI, Comparative Fit Index; RMSEA, root mean square error of approximation; TLI, Tucker-Lewis Index.

their status.[34] From our study, only 6% had not disclosed their status to anyone. HIV status non-disclosure might be a protective behaviour for PLWHA to conceal their status, evade adverse reactions towards themselves, weigh other people's reactions, and as a sign of concern about the implication of their disclosure on their disclosure targets.[70 71] Furthermore, disclosure entails deciding how and to whom to disclose and identifying appropriate opportunities to disclose or devising means to conceal ones' status and/or medication in order to improve access and adherence to their treatment regimen.

The co-occurrence of depressive and anxiety symptoms was positively correlated with overall HIV-related stigma scores, personalised stigma and negative self-image. This finding corroborates previous studies among PLWHA carried out within SSA[26 30 72] and outside,[73 74] which have invariably found a significant association between HIV-related stigma and depressive symptoms. Liu *et al*[75] reported that the more stigma PLWHA perceived, the more anxiety they experienced. Similarly, we report that HIV-related stigma is significantly associated with the co-occurrence of depressive and anxiety symptoms. Additionally, an individual's perception of themselves in light of their diagnosis appears to trigger depression.[76] Screening for depression, anxiety and HIV-related stigma might provide insights on interventions that may promote a positive attitude and self-image, thereby reducing depression, anxiety and stigma, leading to psychological and physical well-being. Given the cross-sectional nature of the study, we cannot claim causality. However, the association between co-occurrence of depressive and anxiety symptoms and stigma provides the impetus for: (1) longitudinal studies to elucidate causal pathways and (2) targeted interventions to address both stigma and mental health to improve health outcomes of adults living with HIV.

Other factors influencing the four subscales were also established. Having no level of education was positively associated with higher reported disclosure concerns and negative self-image, corroborating findings of studies carried out in Nigeria[77] and the USA.[78] Lower levels of education may lead to less exposure, lack of or little knowledge about HIV infection and transmission. In contrast, higher levels of education might lead to higher levels of knowledge, providing exposure to new ways of thinking and new sources of information about the HIV pandemic resulting in the reduction of less supportive attitudes towards PLWHA.[79 80] Previous research has demonstrated that people with high levels of knowledge of the transmission routes for HIV consistently had more supportive attitudes towards those with HIV demonstrating the role that knowledge has in reducing the misconceptions that act to create fear and shape stigma.[79]

Months since HIV diagnosis was inversely associated with *disclosure concerns,* with persons with a more recent diagnosis reporting greater levels of concern about HIV status disclosure. This is consistent with a study of PLWHA in China[81] and among African Americans.[78] This finding suggests that living longer with HIV is associated with positive outcomes because PLWHA are likely to adjust over time to their HIV positive status, receive more information, develop greater insights and understanding of the disease and establish psychological mechanisms to better cope with HIV stigma leading to lower levels of perceived HIV stigma.

### Strengths and limitations of this study

A potential strength is that this is the first study to report the measurement characteristics of the HSS-12 in the SSA context. We recognise several potential limitations in this study. First, the study was in a clinical setting where our study sample consisted of adults living with HIV on cART. Compared with untreated individuals living with HIV, it is likely that levels of HIV stigma would be lower in our sample because it has been shown that access to ART lowers stigma.[82–84] Second, this study is cross-sectional, so causality for the observed significant associations cannot be inferred. We can also not conclude how individuals may experience stigma over time because of the study design limitation. Third, findings may not be generalisable to all adults living with HIV in Kenya as data were collected from one geographical setting and excluded adults

older than 60 years. Fourth, because many participants (n=145) lacked information on their most recent viral load and none had follow-up data on CD4 counts, these variables were excluded from the regression analyses. A disproportionately large number of patients, combined with financial constraints, may explain why these tests are not routinely performed. Future studies, particularly those from resource-constrained settings, should budget for these tests because these biological factors have been associated with HIV-related stigma.[85] Finally, the psychometric robustness of the disclosure concern subscale may be limited. We recommend further research into investigating this specific subscale.

## Conclusions and implications

From the study, the 12-item short version of the Berger HSS[14] had good psychometric properties and can be recommended for research purposes. The current study suggests that women, those who have not disclosed, and those experiencing co-occurring depressive and anxiety symptoms experience a higher level of perceived HIV stigma in Coastal Kenya. This finding is useful in designing future interventions to improve the quality of life of PLWHA. We propose interventions that need to take into account gender to address the specific needs of women, people who have not disclosed their HIV status, and those exhibiting symptoms of depression and anxiety, thereby improving their quality of life. All these interventions will help in bettering both the physical and mental well-being of adults living with HIV. Additionally, it would be prudent to investigate further the association between lower education and HIV-related stigma as we found a marginal association.

**Author affiliations**
[1]Department of Public Health and Primary Care, Faculty of Medicine and Health Sciences Ghent University, Ghent, Belgium
[2]Department of Social Sciences, School of Humanities and Social Sciences Pwani University, Kilifi, Kenya
[3]Department of Clinical Research (Neurosciences), KEMRI/Wellcome Trust Research Programme, Centre for Geographic Medicine Research (Coast), Kilifi, Kenya
[4]Department of Clinical, Neuro- and Developmental Psychology, Amsterdam Public Health Research Institute, Vrije Universiteit Amsterdam, Amsterdam, The Netherlands
[5]Institute for Human Development, The Aga Khan University, Nairobi, Kenya
[6]Department of Epidemiology and Preventive Medicine, Monash University, Melbourne, Victoria, Australia
[7]Department of Psychiatry, Oxford University, Oxford, UK
[8]Department of Public Health, Pwani University, Kilifi, Kenya

**Acknowledgements** We thank our participants for taking the time and effort to participate in this study and the Comprehensive Care and Research Centre (CCRC) staff in the Kilifi County Hospital HIV care and treatment clinic for their unrelenting support. We appreciate Patrick Katana Vidzo for helping with the entry of clinical data. The authors acknowledge the Kenya Medical Research Institute (KEMRI) Director for permission to publish this work.

**Contributors** SWW, CRJCN and AA conceptualised the study. SWW, MKN and AA designed the study. PM formulated study questions for tablet administration and managed the data. SWW and MKN supervised data collection. SWW, MKN, and AMM participated in data collection. SWW and MKN analysed the data. SWW, MKN, PM, AMM, SL, CRJCN and AA contributed to interpreting the data. SWW wrote the first draft of the manuscript. All authors reviewed subsequent versions of the manuscript and approved the final version for submission. The corresponding author affirms that all listed authors meet authorship criteria and that no other author meeting the criteria has been omitted. As guarantor, AA accepts full responsibility for the conduct of the study, had access to the data, and controlled the decision to publish.

**Funding** This work was supported by funding from the Medical Research Council (Grant number MR/M025454/1) to AA. This award is jointly funded by the UK Medical Research Council (MRC) and the UK Department for International Development (DFID) under the MRC/DFID concordant agreement and is also part of the EDCTP2 programme supported by the European Union.

**Disclaimer** The funders had no role in the design, collection, analysis, interpretation and manuscript writing.

**Competing interests** None declared.

**Patient consent for publication** Not applicable.

**Ethics approval** This study involves human participants and was approved by The local institutional review board, Scientific and Ethics Review Board (SERU; Ref KEMRI/SERU/CGMR-C/108/3594), granted ethical approval to recruit participants into the study. We obtained authorisation to work in the HIV care and treatment clinic from the Ministry of Health, County government of Kilifi (Ref HP/KCHS/VOL. VIX/65). Study participants provided written, informed consent to be part of the study. Participants gave informed consent to participate in the study before taking part.

**Provenance and peer review** Not commissioned; externally peer reviewed.

**Data availability statement** Data are available on reasonable request. No additional data are available. Anyone interested in accessing the data reported in this article is free to write to the Data Governance Committee of the KEMRI Wellcome Trust Research Programme, review the application and advise as appropriate, and ensure that uses are compatible with the consent obtained from participants for data collection. Requests can be sent to the coordinator of the Data Governance Committee using the following email: dgc@kemri-wellcome.org.

**ORCID iDs**
Stanley W Wanjala http://orcid.org/0000-0003-1422-0162
Moses K Nyongesa http://orcid.org/0000-0002-7761-0718

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
