## [Reviewer comments · BMJ Open]

ARTICLE DETAILS

TITLE (PROVISIONAL)	Measurement characteristics and correlates of HIV-related stigma among adults living with HIV: A cross-sectional study from coastal Kenya
AUTHORS	Wanjala, Stanley ; Nyongesa, Moses; Mwangi, Paul; Mutua, Agnes M.; Luchters, Stanley; Newton, Charles; Abubakar, Amina A.

VERSION 1 – REVIEW

REVIEWER	Grosso, Ashley Rutgers University Institute for Health Health Care Policy and Aging Research
REVIEW RETURNED	26-Apr-2021

GENERAL COMMENTS	This is an interesting paper that I think could be strengthened by addressing the issues below: Introduction • The first paragraph jumps from global statistics to the percentage of people living with HIV in Kenya are adults without transition and then back to global statistics. It would be helpful to put all the global statistics together and then provide some more relevant information about Kenya, like the total number of people living with HIV in Kenya, the proportion of those who are on ART, number of deaths, etc.• “HIV-related stigma” should be hyphenated consistently. Methods • More details about pretesting would be helpful. How many people were involved? Were they similar to the sample in this study?• Consider using the ordinal alpha based on polychoric correlations instead of or in addition to Cronbach’s alpha to assess internal consistency. Results • There is something wrong with a reference or something (“Supplementary Error! Reference source not found.”) Discussion • Additional limitations should be acknowledged regarding the disclosure concern subscale. On this subscale item 6 had a low factor loading and item-total correlation coefficient, and Cronbach’s alpha was less than acceptable. Did you try to remove item 6 to see if it would improve the psychometric properties of the subscale and overall scale?• This sentence is a little confusing to me and could be reworded for clarity: “Further, disclosure is not only about how or whom to disclose to, but it also entails finding good opportunities to disclose or devise means of keeping ones’ status and/or medication a
---

	secret to enhance access and adherence to their treatment regimen.”  • Consider using a word/term other than “influenced” here to indicate that it is correlation and not necessarily causal: “The co-occurrence of depressive and anxiety symptoms influenced overall HIV related stigma scores” Conclusions and implications  • Consider using “adults living with HIV” rather than “HIV-infected adults”. Table 1  • I think there is an error in the percentage for viral load. It should be 86.9 not 89.6.
--	---

REVIEWER	Dickson-Gomez, J Medical College of Wisconsin, Psychiatry and Behavioral Medicine
REVIEW RETURNED	10-Jun-2021

GENERAL COMMENTS	This is a well written paper that tests the validity of an HIV-stigma scale among a sample of PLWHA in coastal Kenya, and the correlates of HIV-stigma. The results are clearly presented and the discussion is well presented. A minor issue is that in the description of the research setting, the authors do not clearly state that ART is provided in the clinic. The fact that HIV related stigma did not correlate with viral load is somewhat surprising given the fact that has been so highly correlated with HIV morbidity and mortality in other settings. A brief mention of this null finding with an explanation would strengthen the paper. A couple of grammatical errors need correction. The authors write repeated "either separated, divorced or widowed." Either is usually used for a choice between two things, not three. The sentence in the discussion, "Anakwa and colleagues...reported greater levels of HIV disclosure concerns, therefore less likely to disclose their status" is missing a verb after therefore, I think.
---

REVIEWER	Stephens, Jacqueline Flinders University, College of Medicine and Public Health
REVIEW RETURNED	15-Jun-2021

GENERAL COMMENTS	This was a very well written and detailed description of the psychometric properties of the HSS-12 among Kenyan PLWHA. The authors demonstrate the HSS-12 is useful for identifying stigma in this population and identified subsets of the population who experienced greater perceived stigma. I mainly only found minor editorial issues: Introduction Page 4, Line 18: Reference needed at end of sentence “ ... (7.8-26 million).” Page 4, Line 20: The word ‘Million’ does not need to be capitalised. Methods Page 5, Line 52: Remove hyphen after “Medicine Research-“ Page 6, Line 38: Please include the company details for the software used. Page 7, Line 36: The abbreviation ‘CMD’ seems to be first use without explanation.
--

	Page 7, Line 50: What is the maximum score possible for the asset score calculated? Page 9, Line 33: The scores were taken 2 weeks apart. What was the reasoning for this timeframe? Results Page 10, Line 30: Typically starting a sentence with a number should be avoided. Page 10, Line 43: The word 'depressive' should be replaced with 'depression'. Page 10, Line 50: There is a document reference error for Supplementary Table 1. Page 11, Lines 36-60: Could much of the two paragraphs under heading "Correlated of perceived HIV related stigma" be added to (or put in) a table? Page 12, Lines 12 & 38: Please refer to Tables 3 and 4 earlier in the two paragraphs where their data is presented, rather than at the end of the paragraph. Discussion Page 12, Line 50: What do you mean by "borderline statistically significant"? I recommend removing this statement. Page 13, Line 38: This sentence needs a reference after "... been published". Page 14, Line 13: Replace "like ours" with "like in Kenya" Page 14, Line 15: Please be consistent with using terminology: females, males, women, men Page 14, Line 38: Remove 'a' from "a PLWHA". References: Error in reference 1. Tables There is inconsistent bolding of the text throughout all the tables. Table 1: Please make it clear (18-60) is the age range.
--	---

VERSION 1 – AUTHOR RESPONSE

Comments from Reviewer 1: Dr. Ashley Grosso, Rutgers University Institute for Health Care Policy and Aging Research

Comments to the Author:

This is an interesting paper that I think could be strengthened by addressing the issues below:

Author Response: We thank Dr. Grosso for her positive feedback.

Introduction

- The first paragraph jumps from global statistics to the percentage of people living with HIV in Kenya are adults without transition and then back to global statistics. It would be helpful to put all the global statistics together and then provide some more relevant information about Kenya, like the total number of people living with HIV in Kenya, the proportion of those who are on ART, number of deaths, etc.

Author Response: We agree with the reviewer that compiling all global statistics and then providing more relevant information about HIV statistics in Kenya would be extremely beneficial. As a result, we made the suggested changes to the manuscripts' introductory paragraphs, making it easier for the reader to follow through (see page 4 lines 46 to 53).

- "HIV-related stigma" should be hyphenated consistently.

Author Response: We agree with the reviewer's comment and have subsequently hyphenated all mentions of HIV-related stigma in the manuscript for consistency. See page 2 line 9; Page 5 line 85, 86, 91, 93; Page 12 line 293; Page 13 line 332; Page 14 line 371, 380; Page 15 line 392, 396

Methods

- More details about pre-testing would be helpful. How many people were involved? Were they similar to the sample in this study?

Author Response: Many thanks to the reviewer for this comment. We have added a statement that helps to clarify the number of people involved in the pre-testing. "Before conducting the formal phase of the study, fifteen pre-test interviews were conducted to assess instrumentation rigour and develop measures to address any limitations or threats to bias and management procedures." (see page 8 lines 190-194 for this information) In addition, since this data is part of a larger study, we have provided a reference for more information related to Instrument translation and cross-cultural adaptation

Nyongesa MK, Mwangi P, Wanjala SW, et al. Correlates of health-related quality of life among adults receiving combination antiretroviral therapy in coastal Kenya. *Health quality of life outcomes* 2020; 18:1-14. doi: 10.1186/s12955-020-01421-0

- Consider using the ordinal alpha based on polychoric correlations instead of or in addition to Cronbach's alpha to assess internal consistency.

Author Response: Many thanks to the reviewer for suggesting that, in addition to Cronbach's alpha, Ordinal alphas based on polychoric correlations be used to assess internal consistency. We addressed this concern by including ordinal alphas (please see page 9, lines 224; Page 11, lines 286-287; Page 13, lines 343-346), which have been recommended for use with rating scales or Likert-type response formats. However, Chalmers, writing on the misconceptions and the limited usefulness of Ordinal alpha, concludes that ordinal alpha should not be used in routine reliability analyses and reports. Instead, it should be understood as a hypothetical tool, similar to the Spearman-Brown prophecy formula, for theoretically increasing the number of ordinal categorical response options in future applied testing applications. Additionally, he advocates for ordinal alpha to be interpreted as a measure of hypothetical reliability instead of observed reliability. For this reason, we have included the ordinal alphas in the manuscript in addition to the Cronbach's alpha but only interpreted the Cronbach's alphas.

Chalmers RPJE, *Measurement P*. On misconceptions and the limited usefulness of ordinal alpha. 2018;78(6):1056-71. doi: 10.1177/0013164417727036

Results

- There is something wrong with a reference or something ("Supplementary Error! Reference source not found.")

Author Response: This concern has been addressed

Discussion

- Additional limitations should be acknowledged regarding the disclosure concern subscale. On this subscale item 6 had a low factor loading and item-total correlation coefficient, and Cronbach's alpha was less than acceptable. Did you try to remove item 6 to see if it would improve the psychometric properties of the subscale and overall scale?

Author Response: Many thanks to the reviewer for this comment. During the initial analysis, we removed item 6 to see if the psychometric properties of the subscale and overall scale would improve. However, Cronbach's alpha for the overall scale improved from 0.80 to 0.81 and from 0.53 to 0.54 for the disclosure subscale. All the other three subscales remained the same. As a result, we have included it as a limitation in our study and acknowledge that the psychometric robustness of the disclosure subscale may be limited thus, we recommend further research on this subscale. (see page 16 lines 441-443)

- This sentence is a little confusing to me and could be reworded for clarity: "Further, disclosure is not only about how or whom to disclose to, but it also entails finding good opportunities to disclose or devise means of keeping ones' status and/or medication a secret to enhance access and adherence to their treatment regimen."

Author Response: We thank the reviewer for this comment and have reworded it for clarity. The sentence now reads, "Furthermore, disclosure entails deciding how and to whom to disclose as well as identifying appropriate opportunities to disclose or devising means to conceal ones' status and/or medication in order to improve access and adherence to their treatment regimen. (See Page 15 lines 388-390)

- Consider using a word/term other than "influenced" here to indicate that it is correlation and not necessarily causal: "The co-occurrence of depressive and anxiety symptoms influenced overall HIV related stigma scores"

Author Response: We agree with the reviewer and have replaced the word influence with the phrase positively correlated, resulting in the following new sentence: "The co-occurrence of depressive and anxiety symptoms was positively correlated with overall HIV-related stigma."

(Page 15 lines 396-397)

Conclusions and implications

• Consider using “adults living with HIV” rather than “HIV-infected adults”.

Author Response: We have replaced all mentions of “HIV-infected adults” with “adults living with HIV” to use the first-person language consistently (Page 17, line 453-454)

Table 1

• I think there is an error in the percentage for viral load. It should be 86.9 not 89.6.

Author Response: We thank the reviewer for picking this up. We discovered and acknowledged this error after inspecting Table 1 and calculating the percentage further. We have changed it to 86.9. (Page 25)

Comments from Reviewer 2: Prof. J Dickson-Gomez, Medical College of Wisconsin

Comments to the Author:

This is a well written paper that tests the validity of an HIV-stigma scale among a sample of PLWHA in coastal Kenya, and the correlates of HIV-stigma. The results are clearly presented, and the discussion is well presented.

Author Response: We would like to thank the reviewer for the positive appraisal of our manuscript, and we are especially pleased to hear that the results and discussion sections are well presented.

A minor issue is that in the description of the research setting, the authors do not clearly state that ART is provided in the clinic

Author Response: In response to the reviewer’s comment, we have updated the research setting details to include a statement clearly describing that ART is provided in the clinic (page 6, line 108-109).

The fact that HIV related stigma did not correlate with viral load is somewhat surprising given the fact that has been so highly correlated with HIV morbidity and mortality in other settings. A brief mention of this null finding with an explanation would strengthen the paper.

Author Response: We thank the reviewer for this comment. We provide a detailed account of why Viral load was not included in the multivariable analysis; hence there was no correlation with HIV stigma. As captured in the limitations of our study, “....., because many participants (n = 145) lacked information on their most recent viral load and none had follow-up data on CD4 counts, these variables were excluded from the regression analyses. A disproportionately large number of patients, combined with financial constraints, may explain why these tests are not routinely performed. Future studies, particularly those from resource-constrained settings, should budget for these tests because these biological factors have been associated with HIV-related stigma. (see page 16 lines 435-443)

Kemp CG, Lipira LL, David H, et al. HIV stigma and viral load among African-American women receiving treatment for HIV: A longitudinal analysis. 2019;33(9):1511.

A couple of grammatical errors need correction. The authors write repeated “either separated, divorced or widowed.” Either is usually used for a choice between two things, not three.

Author Response: We thank the reviewer for picking this up. We have corrected all spelling, and grammatical errors pointed out by the reviewer.

The sentence in the discussion, “Anakwa and colleagues...reported greater levels of HIV disclosure concerns, therefore less likely to disclose their status,” is missing a verb after therefore, I think.

Author Response: Thanks for the comment. We have added the verb after therefore and it now reads, “Anakwa and colleagues found that PLWHA with higher levels of perceived HIV-related stigma reported greater levels of HIV disclosure concerns; therefore, they are less likely to disclose their status.” Page 14 lines 382-384

Comments from Reviewer 3: Dr Jacqueline Stephens, Flinders University

Comments to the Author:

This was a very well written and detailed description of the psychometric properties of the HSS-12 among Kenyan PLWHA. The authors demonstrate the HSS-12 is useful for identifying stigma in this population and identified subsets of the population who experienced greater perceived stigma. I mainly only found minor editorial issues:

Author Response: We would like to thank the reviewer for this positive appraisal. We are particularly pleased to learn that the reviewer found the manuscript well written and that it provides a detailed description of the psychometric properties of the HSS-12 among Kenyan adults.

Introduction

Page 4, Line 18: Reference needed at end of sentence "... (7.8-26 million)."

Author Response: We have included a reference at the end of the sentence as suggested by the reviewer. The sentence now reads as follows (Page 4, lines 45-47):

"Between 2010 and mid-2020, there has been an upsurge in the number of people accessing antiretroviral therapy (7.8- 26 million)."

Joint United Nations Programme on HIV/AIDS. Global HIV & AIDS statistics — 2020 fact sheet [Available from: <https://www.unaids.org/en/resources/fact-sheet> accessed 22/2/2021.

Page 4, Line 20: The word 'Million' does not need to be capitalised.

Author Response: Many thanks for this comment. We have now removed the capital letter. (Page 4 line 48)

Methods

Page 5, Line 52: Remove hyphen after "Medicine Research- "

Author Response: We thank the reviewer for this comment. We have removed the hyphen (please see page 5, line 101).

Page 6, Line 38: Please include the company details for the software used.

Author Response: We thank the reviewer for this suggestion. We now cite the company details for all the software used. (Please see the added citations below and on pages 6 line 126; Page 10 line 248 and line 250 of the manuscript).

REDCap:

Harris PA, Taylor R, Minor BL, et al. The REDCap consortium: Building an international community of software platform partners. 2019;95:103208.

Harris PA, Taylor R, Thielke R, et al. Research electronic data capture (REDCap)—a metadata-driven methodology and workflow process for providing translational research informatics support. 2009;42(2):377-81.

Stata:

StataCorp LP. Stata statistical software: release 14. [computer program] College Station, TX; StataCorp LP: 2015.

R studio:

R Core Team. R: A language and environment for statistical computing. R Foundation for Statistical Computing. [computer program] 4.0.2 Vienna, Austria. 2020.

Page 7, Line 36: The abbreviation 'CMD' seems to be first use without explanation.

Author Response: We thank the reviewer for this comment. On page 7, line 161, we have addressed this concern to indicate that CMD (symptoms of common mental disorders) indicates the co-occurrence of depressive and anxiety symptoms

Page 7, Line 50: What is the maximum score possible for the asset score calculated?

Author Response: The asset index score comprises seven items enquiring on disposable assets owned by the participants. The maximum possible score from the asset index is seven (7). This is now indicated on page 8, line 170-171.

Page 9, Line 33: The scores were taken 2 weeks apart. What was the reasoning for this timeframe?

Author Response: We thank the reviewer for this question. To clarify, previous work has shown that the period between the repeated administrations should be long enough to prevent recall, though short enough to ensure that clinical change has not occurred. Often, 1 or 2 weeks will be appropriate, but there could be reasons to choose otherwise.

Terwee CB, Bot SD, de Boer MR, et al. Quality criteria were proposed for measurement properties of health status questionnaires. 2007;60(1):34-42.

Results

Page 10, Line 30: Typically starting a sentence with a number should be avoided.

Author Response: We thank the reviewer for this comment. We have revised the sentence to avoid starting with a number, and it now reads, "A total of 417(93.7%) were in stage 1 of the WHO clinical staging, and 425 (95.3%) were on the first-line cART regimen (Table 1)." Page 10 lines 259-261

Page 10, Line 43: The word 'depressive' should be replaced with 'depression'.

Author Response: On page 11, line 267, we have replaced the word 'depressive' with 'depression' as suggested by the reviewer.

Page 10, Line 50: There is a document reference error for Supplementary Table 1.

Author Response: We thank the reviewer for noticing this. The reference error for supplementary Table 1 has now been rectified. The problem arose because the supplemental Table 1 had not been included in the main document.

Page 11, Lines 36-60: Could much of the two paragraphs under heading "Correlated of

perceived HIV related stigma” be added to (or put in) a table?

Author Response: We thank the reviewer for this comment. All the information that the reviewer refers to is included in Table 3, which deals with univariate linear regression of correlates of perceived HIV-related stigma among adults living with HIV from rural Kilifi.

Page 12, Lines 12 & 38: Please refer to Tables 3 and 4 earlier in the two paragraphs where their data is presented, rather than at the end of the paragraph.

Author Response: Thank you for this comment. As suggested, we now refer to Tables 3 and 4 at the beginning of the paragraph than at the end. (See Page 12 lines 294-295)

Discussion

Page 12, Line 50: What do you mean by “borderline statistically significant”? I recommend removing this statement.

Author Response: We thank the reviewer for this comment. We have removed the phrase borderline statistically significant and replaced it with “...having no education was associated with increasing stigma levels at $p=0.051$ ($\beta=3.318$, 95%CI: $-.01$, 6.65).” Page 12 lines 317-318

Page 13, Line 38: This sentence needs a reference after “... been published”.

Author Response: We have now added a reference that corroborates our findings. (Page 14 line 356)

Page 14, Line 13: Replace “like ours” with “like in Kenya”

Author Response: We thank the reviewer for this suggestion. On Page 14, line 372, we have amended the phrase “like ours” to read “like Kenya.”

Page 14, Line 15: Please be consistent with using terminology: females, males, women, men

Author Response: We thank the reviewer for picking up this inconsistency. Throughout the manuscript, we now consistently use the terms ‘female’ and ‘male.’

Page 14, Line 38: Remove ‘a’ from “a PLWHA”.

Author Response: Thank you for this comment. We have removed the article ‘a’ as suggested by the reviewer

References:

Error in reference 1.

Author Response: We have made the necessary amendments to Reference 1.

Joint United Nations Programme on HIV/AIDS. Global HIV & AIDS statistics — 2020 fact sheet [Available from: <https://www.unaids.org/en/resources/fact-sheet> accessed 22/2/2021.

Tables

There is inconsistent bolding of the text throughout all the tables.

Author Response: We thank the reviewer for noticing this. We have subsequently made all necessary amendments to the tables. We have used an asterisk (*) to indicate the significance of P values and have included the explanations in the table footnotes.

Table 1: Please make it clear (18-60) is the age range.

Author Response: We thank the reviewer for this comment. On page 6, line 116, we have made it clear that the age range is (18-60).

VERSION 2 – REVIEW

REVIEWER	Grosso, Ashley Rutgers University Institute for Health Health Care Policy and Aging Research
REVIEW RETURNED	30-Nov-2021

GENERAL COMMENTS	The first paragraph of the introduction is now more structured but could possibly benefit from some transition sentences. For example, after talking about the disease burden there is a sudden jump to mentioning more positive trends like antiretroviral therapy numbers, and declines in new infections and deaths. Perhaps the authors could add something like “Despite this burden...” Also it is a little unclear if the numbers related to ART, new infections, and deaths are global or for SSA specifically. Similarly, the paragraph jumps to talking about Kenya. Perhaps this should be in a new
--

	paragraph, or there could be a transition sentence indicating why Kenya is important, reflects the larger trends in Africa, etc. I thank the authors for providing the interesting information and reference about ordinal alphas.
REVIEWER	Stephens, Jacqueline Flinders University, College of Medicine and Public Health
REVIEW RETURNED	24-Sep-2021
GENERAL COMMENTS	My thanks to the authors for addressing my queries, as well as addressing the queries of the editor and other reviewers. I have no further queries in regard to this manuscript.